# Methods detecting rhythmic gene expression are biologically relevant only for strong signal

**David Laloum**[1,2], **Marc Robinson-Rechavi**[1,2]*

**1** Department of Ecology and Evolution, Batiment Biophore, Quartier UNIL-Sorge, Université de Lausanne, Lausanne, Switzerland, **2** Swiss Institute of Bioinformatics, Batiment Génopode, Quartier UNIL-Sorge, Université de Lausanne, Lausanne, Switzerland

* marc.robinson-rechavi@unil.ch

**Data Availability Statement:** Data are publicly available from the NCBI GEO database, as specified in the Materials/Datasets section.

## Abstract

The nycthemeral transcriptome embodies all genes displaying a rhythmic variation of their mRNAs periodically every 24 hours, including but not restricted to circadian genes. In this study, we show that the nycthemeral rhythmicity at the gene expression level is biologically functional and that this functionality is more conserved between orthologous genes than between random genes. We used this conservation of the rhythmic expression to assess the ability of seven methods (ARSER, Lomb Scargle, RAIN, JTK, empirical-JTK, GeneCycle, and meta2d) to detect rhythmic signal in gene expression. We have contrasted them to a naive method, not based on rhythmic parameters. By taking into account the tissue-specificity of rhythmic gene expression and different species comparisons, we show that no method is strongly favored. The results show that these methods designed for rhythm detection, in addition to having quite similar performances, are consistent only among genes with a strong rhythm signal. Rhythmic genes defined with a standard $p$-value threshold of 0.01 for instance, could include genes whose rhythmicity is biologically irrelevant. Although these results were dependent on the datasets used and the evolutionary distance between the species compared, we call for caution about the results of studies reporting or using large sets of rhythmic genes. Furthermore, given the analysis of the behaviors of the methods on real and randomized data, we recommend using primarily ARS, empJTK, or GeneCycle, which verify expectations of a classical distribution of $p$-values. Experimental design should also take into account the circumstances under which the methods seem more efficient, such as giving priority to biological replicates over the number of time-points, or to the number of time-points over the quality of the technique (microarray vs RNAseq). GeneCycle, and to a lesser extent empirical-JTK, might be the most robust method when applied to weakly informative datasets. Finally, our analyzes suggest that rhythmic genes are mainly highly expressed genes.

## Author summary

To be active, genes have to be transcribed to RNA. For some genes, the transcription rate follows a circadian rhythm with a periodicity of approximately 24 hours; we call these genes "rhythmic". In this study, we compared methods designed to detect rhythmic genes

**Funding:** Funding was received from Schweizerischer Nationalfonds zur Förderung der Wissenschaftlichen Forschung (173048, to MR-R). The funders had no role in study design, data collection and analysis, decision to publish, or preparation of the manuscript.

**Competing interests:** The authors have declared that no competing interests exist.

in gene expression data. The data are measures of the number of RNA molecules for each gene, given at several time-points, usually spaced 2 to 4 hours, over one or several periods of 24 hours. There are many such methods, but it is not known which ones work best to detect genes whose rhythmic expression is biologically functional. We compared these methods using a reference group of evolutionarily conserved rhythmic genes. We compared data from baboon, mouse, rat, zebrafish, fly, and mosquitoes. Surprisingly, no method was particularly effective. Furthermore, we found that only very strong rhythmic signals were relevant with each method. More precisely, when we use a usual cut-off to define rhythmic genes, the group of genes considered as rhythmic contains many genes whose rhythmicity cannot be confirmed to be biologically relevant. We also show that rhythmic genes mainly contain highly expressed genes. Finally, based on our results, we provide recommendations on which methods to use and how, and suggestions for future experimental designs.

This is a *PLOS Computational Biology* Benchmarking paper.

## Introduction

The nycthemeral transcriptome is characterized by the set of genes that display a rhythmic change in their mRNAs levels with a periodicity of 24 hours. These include, but are not limited to, circadian genes whose rhythm is endogenous and entrainable. In baboon, 82% of protein-coding genes have been reported to be rhythmic in at least one tissue [1]. The nycthemeral rhythmicity of these transcripts can be driven by the internal oscillator clock or by other circadian input such as food-intake, the light-dark cycle, sleep-wake behavior, or social activities. Moreover, the nycthemeral transcriptome is tissue-specific [2, 3]. Given the importance of biological rhythms in understanding biology and medicine, many algorithms have been proposed to detect such rhythms. Some were developed specifically for biological data, while others were adapted from other fields where periodicity is important, such as Lomb Scargle (LS). Most methods are based on non-parametric models that search for referenced patterns, classically sinusoid, called time-domain methods, while some are frequency-domain methods based on spectral analysis [4]. Some of them have been designed to detect more diverse waveforms, including asymmetric patterns, such as RAIN [5] or empirical_JTK (empJTK) [6]. For instance, RAIN outperformed the original JTK_CYCLE algorithm for simulated data consisting of sinusoidal and ramp waveforms [6]. Thus, methods differ in the conception of their algorithm and in how they take into account features of the dataset such as curve shapes, period, noise level, presence of missing data, phase shifts, sampling rates [7], asymmetry of the waveform, or the number of cycles (total period length of the experiment). Each method has in principle different strengths and weaknesses for some features of the dataset. In *Arabidopsis*, HAYSTACK identified 45% more cycling transcripts than COSOPT, mainly due to the inclusion of a 'spike' pattern in its model [8]. Deckard et al. [7] studied how four methods (LS, JTK_CYCLE, de Lichtenberg, and persistent homology) performed across a variety of organisms and periodic processes. Based on synthetic data, they investigated the algorithms' ability to distinguish periodic from non-periodic profiles, to recover period, phase and amplitude, and they evaluated their performance for different signal shapes, noise levels, and sampling

rates. They proposed a decision tree to recommend one of these four algorithms based on these features of datasets [7].

The performance of algorithms to identify such periodic signal has been assessed so far based on synthetic (i.e., simulated) data, or on benchmark sets of known cycling genes. Hughes et al. [9] recently published guidelines for the analysis of biological rhythms and proposed a web-based application (CircaInSilico) for generating synthetic genome biology data to benchmark statistical algorithms for studying biological rhythms. While such benchmarks are useful to explore the behavior of methods in a set of cases, the applicability of results to real data is limited. For example, simulations need to impose an a priori fluctuation pattern, typically cosine. The fluctuation of transcript abundance of core clock genes does seem to follow a cosine shape [10], but sometimes follows non-sinusoidal periodic patterns in mouse liver (e.g., Nr1d1 or Arntl) [11] (based on the data from [12]). The fluctuations of the nycthemeral transcriptome are entrained by a complex network involving external cues [13–16], as simplified in Fig 1a, which might yield non-sinusoidal periodic patterns among rhythmic genes even if

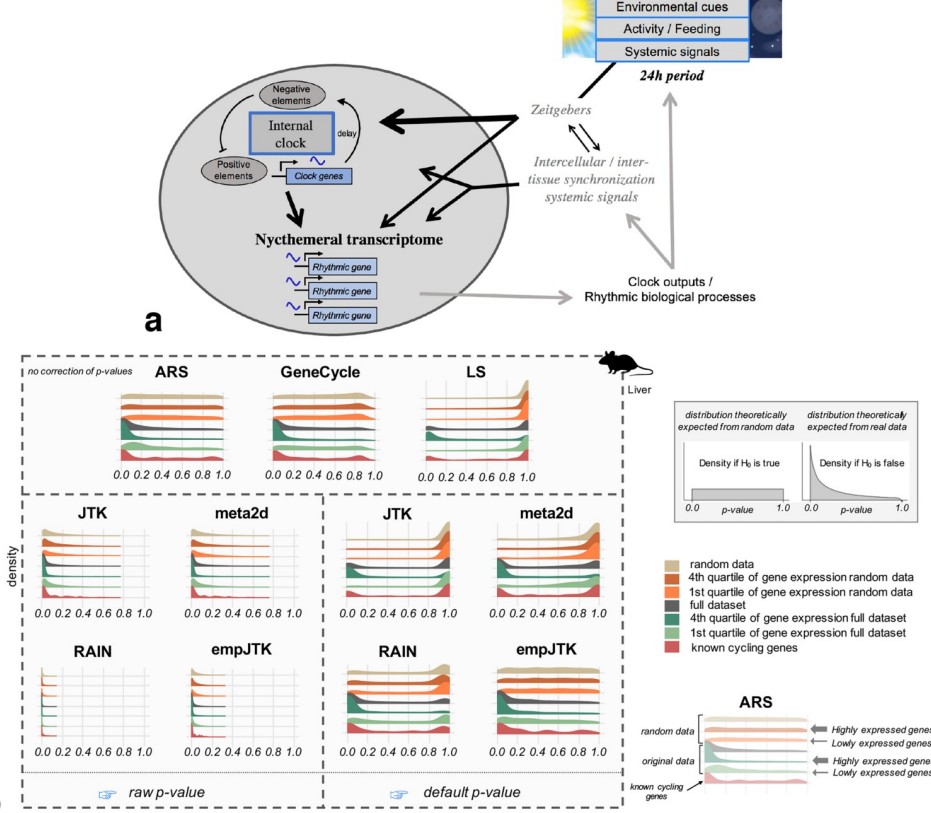

**Fig 1. The nycthemeral transcriptome is the group of genes whose mRNAs have periodic variations with a 24h period, called rhythmic genes. To detect these rhythmic genes, we applied seven methods to time-series datasets that produced different density distribution of *p*-values. a)** Simplified diagram of the entrainment of nycthemeral gene expression. Environmental cues include the light-dark cycle, food-intake, sleep-wake behavior, social activities, or any other 24h periodic event. **b)** Density distribution of *p*-values obtained before (raw) and after the default correction (software) for the seven methods applied to mouse liver data (microarray) sub-categorized in: i. randomized data which represents the null hypothesis; ii. randomized data restricted to the first and fourth quartiles of the median gene expression level, to check for the impact of expression level under the null; iii. the full original dataset; iv. the first and fourth quartiles of the median gene expression level of the original data; and v. a subset of known cycling genes (99 genes from KEGG "circadian entrainment" among which we expect a large proportion of rhythmic mRNA accumulation). The default *p*-values of ARS, GeneCycle, and LS are uncorrected. Mouse image credit to Anthony Caravaggi (license CC BY-NC-SA 3.0).

circadian genes were sinusoidal. But the biological relevance of these waveforms is still not clear. This raises two issues: benchmarks based on simulations are biased towards methods that detect the same types of patterns as simulated; and when an algorithm detects more rhythmic genes, it could be more true positives or more false positives. When pattern constraints are released this increases the number of genes detected as rhythmic, but is not necessarily informative on the capacity of the algorithm to detect genes whose rhythmicity is biologically relevant.

Using real data and randomization tests, we compared seven methods: JTK_CYCLE (**JTK**) [17], **LS** [18, 19], ARSER (**ARS**) [4], and **meta2d** (Fisher integration of LS, ARS, and JTK), are frequently used by many studies and are all included within the MetaCycle R package [20]. We also included empirical_JTK (**empJTK**) [6] and **RAIN** [5], which have been recently developed to deal with more non cosine patterns and with asymmetric waveforms. empJTK and RAIN aim to improve the original JTK algorithm which assumed that any underlying rhythms have symmetric waveforms (more precisely, only the waveform coded into the JTK algorithm will be detected, which is the sine curve by default) [5]. Finally, robust.spectrum [21] extents a robust rank-based spectral estimator to the detection of periodic signals. It is integrated in the GeneCycle R package [22] and called **GeneCycle** in this paper. We excluded de Lichtenberg [23], Persistent Homology [24], COSOPT [25], Fisher's G test [26], MAPES, Capon, and other algorithms for reasons such as i. difficult accessibility of the software which limit their use by researchers, ii. their higher sensitivity to certain features of the data such as the sampling density, the number of replicates and/or periods, noise level, and waveform, iii. their weaker efficiency on simulated data or known cycling genes, or iv. their previously reported less good detection of non-sinusoidal periodic patterns [4, 6, 7, 27–29]. We first analysed the behavior of these seven methods applied to a variety of real datasets in animals, and within each dataset, we compared results between representative gene subsets such as highly and lowly expressed genes, known cycling genes, and randomized data. Contrary to real data, randomized data is not expected to show any signal of rhythmicity, which we used to test proper statistical behavior under the null hypothesis. Secondly, as function tends to be conserved between orthologs [30], true rhythmic genes are expected to be enriched in orthologs that are themselves rhythmic in other species. Indeed, evolutionary conservation provides a valuable filter through which to highlight functional biological networks, notably for clock-controlled functions [31]. The biological relevance of rhythmic genes is expected to be higher for rhythmic orthologs. An unknown proportion of the genes reported as rhythmic but not conserved will be true positives, whose rhythmicity evolved recently in one species or was lost in the other. This would only be a problem if a method would somehow favour these non-conserved ones while reporting true positives; we do not see any reason to expect such a behavior. On the other hand, errors in the prediction of rhythmicity by each method are not expected to be conserved between orthologs. Rather than benchmarking rhythm detection methods based on a profile, we used the biological relevance of genes detected rhythmic. Notably we considered that, among orthologs, those which conserved their rhythmic expression can formed a suitable reference group of rhythmic genes. Thus, the best methods are expected to report rhythmic genes with a high proportion of rhythmic orthologs. We used this approach to compare the algorithms based on their ability to capture biologically relevant evolutionary conservation signal within nycthemeral genes, and compared them to a Naive method.

## Results

We used gene expression time-series datasets that come from circadian experiments and kept the data from healthy, wild-type individuals for seven species (S1 Table), allowing comparisons

among vertebrates and among insects. We benchmarked methods on animal data since organ homology allowed to compare datasets for which we expect conservation of functional patterns (tissue-specific rhythms). For readability, we present vertebrate results in the main figures and insect results in supplementary results (S6 File). Apart from the rat and Anopheles datasets, data with several biological replicates were obtained already normalized over replicates (one value per time-point).

We define a rhythmic gene as a gene which displays a nycthemeral change in its mRNA abundance, i.e. occurring over 24 hours and repeated every 24 hours. All these rhythmic genes represent the nycthemeral transcriptome. Different organs have been reported to have transcriptomes which are more or less rhythmic [2]. The rhythmic expression of these genes can be entrained directly by the internal clock or indirectly by external inputs, such as the light-dark cycle or food-intake [13–16] (Fig 1a). We consider the entirety of these rhythms to be a biologically relevant signal to detect. That is why we preferred data from light-dark and ad-libitum experimental conditions whenever possible (S1 Table), as providing a better representation of wild conditions.

Some methods are distinguished by their higher sensitivity to alternative patterns such as peak, box, or asymmetric profiles. A visual inspection of the KEGG "Circadian entrainment" gene set (see Methods) provides indeed informal confirmation that such patterns can be observed among known cycling genes, such as Npas2, Nr1d1, or Bhlhe41 (S1 File).

## Analysis of statistical behaviors of methods applied to real data

**p-values distribution analysis.**   First, a good method should produce a uniform distribution of p-values when there is no structure in the data, in contrast to the distribution obtained from empirical data, which is expected to be skewed towards low p-values because of the presence of rhythmic genes. We investigated the properties of the different methods applied to randomized vs real data. We also investigated to what extent the density distribution of p-values of each method was affected by gene expression levels. Indeed, higher expression provides more power for detecting rhythmic patterns—highly expressed genes have more chance to shape rhythmic patterns because the variations of expression levels are relative to the general expression level—but this should not be the main driver of results. I.e., a method to detect rhythmicity should not be essentially reporting high expression levels. Even if true rhythmic genes were enriched in high expressed genes, we expect a good method to report both high and low p-values, at each expression level.

Fig 1b shows the density distribution of raw p-values obtained for the seven methods applied to mouse liver data (microarray) sub-categorized in: i. randomized data which represents the null hypothesis; ii. randomized data restricted to the first and fourth quartiles of the median gene expression level, to check for the impact of expression level under the null; iii. the full original dataset; iv. the first and fourth quartiles of the median gene expression level of the original data; and v. a subset of known cycling genes (8 to 99 genes according to species, see Methods). Results from the other datasets are provided in S2 and S3 Files. Surprisingly, only ARS and GeneCycle displayed close to the expected uniform raw p-value distribution for randomized data (Fig 1b). The adjustment by default of empJTK (minimum of the p-value calculated from an empirical null distribution, and of Bonferroni) recovered the expected uniform distribution, suggesting that this correction allows recovering proper p-values (Fig 1b). We used each software output "p-values" for calls, which we call "default p-value". In some software, these values result from an internal p-value adjustment, so we also analysed "raw p-values" (uncorrected, see Methods and Table 1 for JTK). For ARS, GeneCycle, and LS, the default p-values are uncorrected. Under the null hypothesis, LS has an abnormal peak near p-value = 1

**Table 1. Raw, default, and BH.Q in JTK algorithm.**

| JTK | description | R |
|---|---|---|
| raw *p*-value | No correction | - |
| default *p*-value | Bonferroni correction of raw *p*-values | p.adjust(raw.pvals, method="bonf") |
| BH.Q (this paper) | Benjamini-Hochberg correction of raw *p*-values | p.adjust(raw.pvals, method="BH") |
| BH.Q (software) | Benjamini-Hochberg correction of default *p*-values | p.adjust(default.pvals, method="BH") |

(Fig 1b), implying an issue with its definition of the null hypothesis, or maybe a one-sided test when a two-sided test would be appropriate. The three other algorithms (RAIN, JTK, and meta2d) seem to have issues with false positives, displaying large proportions of low *p*-values even for randomized data. This issue was also recently reported by Hutchison and Dinner [32] who in addition showed that a combined method, such as meta2d which integrates results from ARS, JTK, and LS, under-perform the individual methods for low *p*-values [32].

Before analysing the impact of expression levels, we checked that the data follow a typical bimodal density distribution of gene expression (S1 File) and that using the median of time-points for gene expression gives similar results to using the minimum or the mean value (S1 File). Unsurprisingly, higher expression levels imply a higher power to detect rhythmic patterns (S1 File). The *p*-values distributions imply that most methods detect almost all highly expressed genes, and almost no lowly expressed genes, as "rhythmic" (Fig 1b). The normalization of gene expression values (Z-score transformation) did not change the *p*-values distributions within highly expressed genes, and particularly did not recover rhythmicity within lowly expressed genes (S1 File). This was not due to sampling biases of microarray data since results are consistent with RNAseq data (S1 File). Thus, the differences obtained between highly and lowly expressed genes either reflect true biology or a lower signal to noise ratio in lowly expressed genes. We think that (i) a method which is able to detect at least some lowly expressed genes as rhythmic is preferable, and (ii) a method should not detect almost all highly expressed genes as rhythmic. Overall, ARS, empJTK, and GeneCycle had the best behavior, producing a uniform distribution under the null hypothesis, and a skew towards low *p*-values for all empirical data.

Much more rhythmic signal is detected among genes with high amplitude (S1 File). This does not necessarily imply that the rhythmicity of the low amplitude genes isn't biologically relevant. From data of the same mouse experiment [2], we observed differences of *p*-value density distributions between microarray and RNAseq, with the skew towards low *p*-values less marked for RNAseq data (S1 File). This can be due to the more precise temporal resolution of the microarray time-series dataset, or to differences in the detection of gene expression by RNAseq vs microarrays (Fig 2a). When we restricted the microarray time-series to the same time-points as in the RNAseq series, we obtained a *p*-value distribution very similar to that of the RNAseq data (Fig 2b). The same time-series restriction applied to known cycling genes produced comparable results (S1 File). This supports a major role of the temporal resolution for method results, relative to a minor role for the difference between RNAseq and microarrays. That is why for the next steps, we only considered the microarray dataset for the mouse.

This observation can be generalized to diverse datasets. We see that each method loses in efficiency when the number of 24h cycles decreases, or when the number of time-points sampled decreases (Fig 3a). We show only results of this comparison for ARS, GeneCycle, and empJTK because they were the only methods with correct behavior in their *p*-value distributions (Fig 1b). For the same number of time-points, performance seems better with two cycles than only one cycle, as shown comparing zebrafish and baboon data which have both twelve time-points (Fig 3a). But this observation could be confused by the comparison of different species or different samples' quality. ARS performed better with a smaller total number of

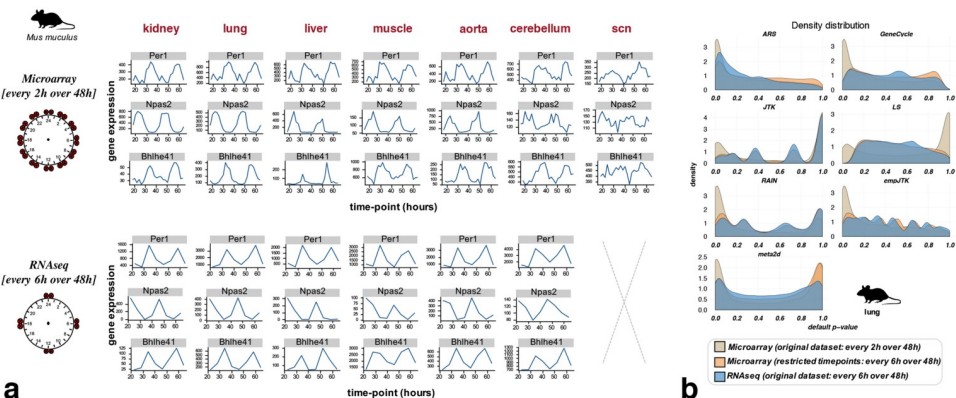

**Fig 2. Fewer time points per cycle lead to a weaker detection of rhythmic patterns even if the transcriptome profiling quality is better. a)** Bhlhe41, Npas2, and Per1 expression over time from data of the same mouse experiment [2] using two transcriptome profiling techniques: microarray vs RNAseq. The number of time-points with data is 24 for microarray and 8 for RNAseq. **b)** The restriction of microarray time-series to the same time-points as in the RNAseq series produces similar *p*-value distributions to those obtained with RNAseq. This supports a major role of the temporal resolution for method results, relative to a minor role for the difference between RNAseq and microarrays. Mouse image credit to Anthony Caravaggi (license CC BY-NC-SA 3.0).

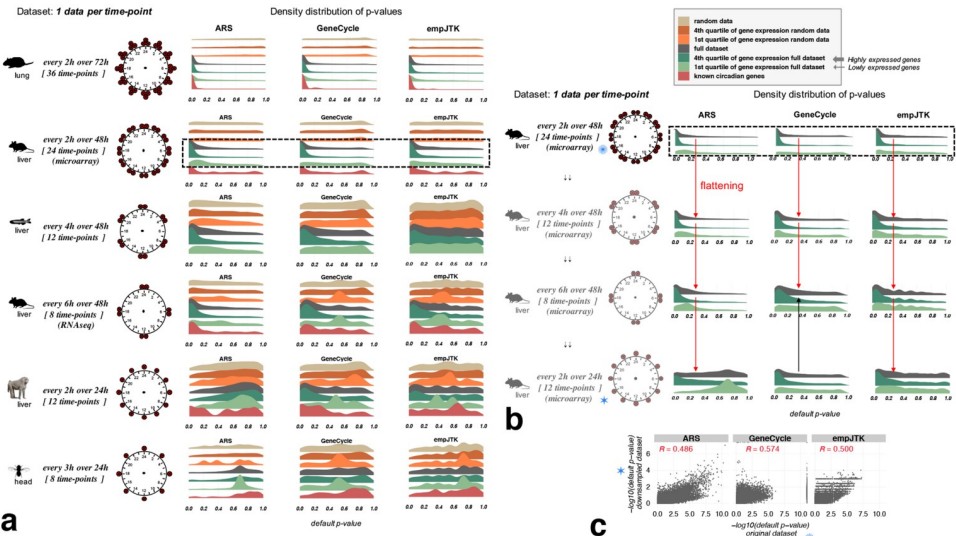

**Fig 3. Datasets with one replicate per time-point over a unique cycle of 24 hours do not provide enough information to detect rhythmicity.** Methods lose in statistical power for detecting rhythmic patterns in gene expression when the number of 24h cycles decreases, or when the number of time-points sampled decreases. **a)** Default *p*-value distributions obtained for ARS, GeneCycle, and empJTK applied to different datasets and sub-categorized in: i. randomized data which represents the null hypothesis; ii. randomized data restricted to the first and fourth quartiles of the median gene expression level, to check for the impact of expression level under the null; iii. the full original dataset; iv. the first and fourth quartiles of the median gene expression level of the original data; and v. a subset of known cycling genes (8 to 99 genes according to species, see Methods). For each dataset, the number of time-points with data and the temporal resolution is illustrated around a 24h clock. For the same number of time-points, performance seems better with two cycles than only one cycle (zebrafish vs baboon). **b)** The reduction of the number of time-points of the mouse liver microarray dataset shows increasingly weak rhythm detection by ARS, GeneCycle, and empJTK, shown by a flattening of the *p*-value distribution on the full dataset (red arrow). GeneCycle showed no difference between a few time-points over two cycles or more time-points over a single cycle (black arrow). **c)** Scatter-plots of *p*-values obtained before and after down-sampling (every 2h over 48h vs. every 2h over 24h) for the full dataset. Each point is a gene. *R* is the Pearson correlation; *p*-value < 2.2e−16 in all cases. After down-sampling, the rhythmic signal is retrieved for the same genes. Images credit: Anthony Caravaggi (mouse), Ian Quigley (zebrafish), wikipedia GNU GPL Muhammad Mahdi Karim (baboon), and Public Domain for other images (from PhyloPic).

time-points but over two cycles than with more total time-points over a single cycle (mouse RNAseq vs baboon in Fig 3a), indicating that ARS is very dependant on the repetitive nature of profiles. The reduction of the number of time-points of the mouse microarray dataset shows similar effects on the rhythm detection by ARS, GeneCycle, and empJTK (Fig 3b). Of note, GeneCycle presented more or less no differences between having a few time-points over two cycles and having more time-points over a single cycle (black arrow Fig 3b).

**Overlap between methods.** Among genes called rhythmic, we analysed the number of those called in common by the different methods. For *p*-value thresholds of 0.05 or 0.01, we found a large proportion of genes called rhythmic by only one or few methods (Fig 4a and S1 File which shows the Jaccard index heatmap for mouse liver). Nevertheless, the overlap between all methods was the largest category for the mouse liver data (Fig 4a). Using a very low false positive tolerance with FDR thresholds of 0.5%, all methods except LS overlap largely (S1 File). If we ignore *p*-value thresholds and consider the first 6000 genes detected rhythmic for each method, the overlap becomes stronger (Fig 4d). We obtained similar results from the most informative dataset (S1 File). Indeed, the rat lung dataset has 36 time-points spread over

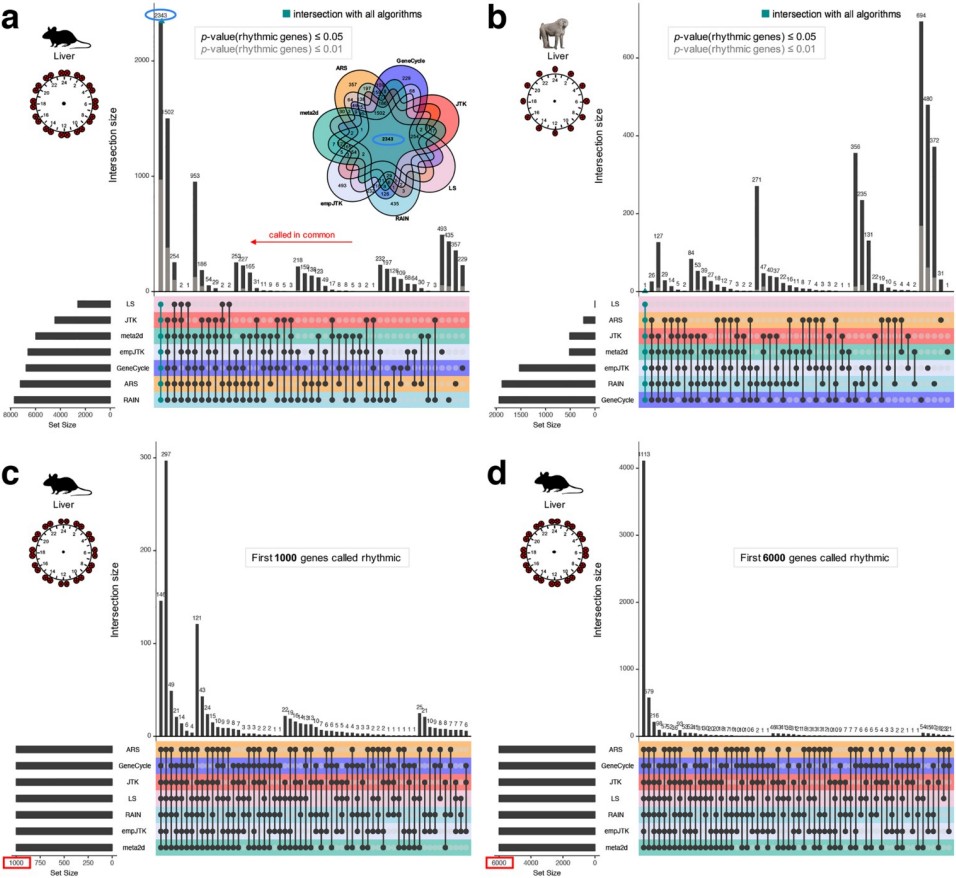

**Fig 4. Methods detect the same first top rhythmic genes, but with inconsistencies in the meaning of their *p*-values.** Upset diagrams show the number of rhythmic genes called in common by the methods. Each intersection is exclusive, i.e. one gene can appear in only one intersection. **(a,b)** Upset diagram for mouse liver dataset (microarray) **(a)** and baboon liver dataset **(b)** for the *p*-value thresholds of 0.05 (black) or 0.01 (grey) for calling genes rhythmic. The Venn diagram **(a)** illustrates the upset diagram with, for instance, 2343 genes called rhythmic by all methods. **(c,d)** Upset diagram for mouse liver dataset (microarray) for the first 1000 **(c)** or 6000 **(d)** genes detected rhythmic for each method. With a smaller number of top rhythmic genes, the overlap between methods is weaker. Images credit: Anthony Caravaggi (mouse) and wikipedia GNU GPL Muhammad Mahdi Karim (baboon).

three 24h cycles (Fig 3a). Thus, the same genes seem to be called rhythmic by all methods but the threshold of significance appears inconsistent. Some methods are expected to produce different *p*-values because their underlying assumptions are different, i.e. other than sinusoidal for RAIN and empJTK. But the bulk of the methods are designed to find sinusoidal patterns and thus should ideally produce similar *p*-values, or at least similar ordering of results. Thus, our observations suggest an issue with the significance of *p*-value thresholds for the methods. While in principle effect size is often more relevant than *p*-value, these methods are all used in practice to produce *p*-values, define a threshold, and provide a list of "rhythmic genes", thus consistency of these *p*-values is important. With a smaller number of top rhythmic genes, the overlap between methods was weaker (Fig 4c and 4d). Thus the methods agree on a large number of rhythmic genes, but not necessarily on the order of significance among them. Finally, for baboon liver data there was less overlap of methods (Fig 4b; S1 File), which might be due to the low information in that data (Fig 3a).

## Use of evolutionary conservation as a benchmark

**Signal of evolutionary conservation.**    We expect biologically relevant rhythmic activity of genes to be more conserved between species than putative false positives from detection methods. For each condition (species and tissue), we defined the group of genes whose orthologs are called rhythmic in the homologous tissue of another species (Fig 5a). For example, starting with all mouse genes, we only kept mouse-zebrafish one-to-one orthologs. Considering the liver, these orthologs were separated into two groups: genes for which the ortholog is detected as rhythmic in zebrafish liver, called rhythmic orthologs; and the remaining one-to-one orthologs (Fig 5b). Mouse-zebrafish orthologs, that are detected rhythmic in zebrafish liver, were significantly more enriched in small *p*-values in mouse liver, for all methods (Kolmogorov-Smirnov test *p*-values < 0.001 with Kolmogorov's D statistic around 10-15% of maximum deviation, Fig 5d). Similar results were obtained using different methods and/or a different threshold to call orthologs as rhythmic in zebrafish liver (S1 File). This result obtained for distant species (S1 File) shows that the conservation of rhythmicity at the transcriptomic level is informative. Similar results were obtained in other species comparisons (S1 File), with a stronger signal for evolutionarily close species such as mouse and rat (with Kolmogorov's D statistic around 10-15% of maximum deviation, S1 File), although we found no consistent correlations of the orthologs *p*-values between the rat and the mouse (S1 File). Of note, the comparison of species under different conditions (light-dark versus dark-dark) is a limitation in itself since the overlap of the rhythmic transcriptome between these two conditions has been shown to be low [33–35] (although this interpretation remains limited by the thresholds used). However, we found a good correlation of *p*-values obtained between these two conditions in the head of *Anopheles gambiae* (*R*=0.605, S1 File) suggesting that this limitation does not hide most of the conserved signal. Thus, for the same homologous organ, rhythmic orthologs have a stronger statistical signal of rhythmicity than non-rhythmic orthologs. We are going to use this evolutionary conservation of the rhythmicity of gene expression in order to compare the performance of methods. We expect that a method which detects more genes with biologically relevant rhythmicity should also detect more conservation of rhythmicity. This is both justified in principle, because evolutionary conservation implies relevance to the functioning of the organism, and in practice, since orthologs of rhythmic genes have smaller *p*-values (Fig 5d).

**Only strong rhythmic signals of gene expression are relevant.**    In this last part, we compared the performances of methods to detect the rhythmic orthologs. For a given dataset, the best method is expected to report rhythmic genes with the highest proportion of rhythmic orthologs. It should be noted that this does not imply that we expect all rhythmic behavior to

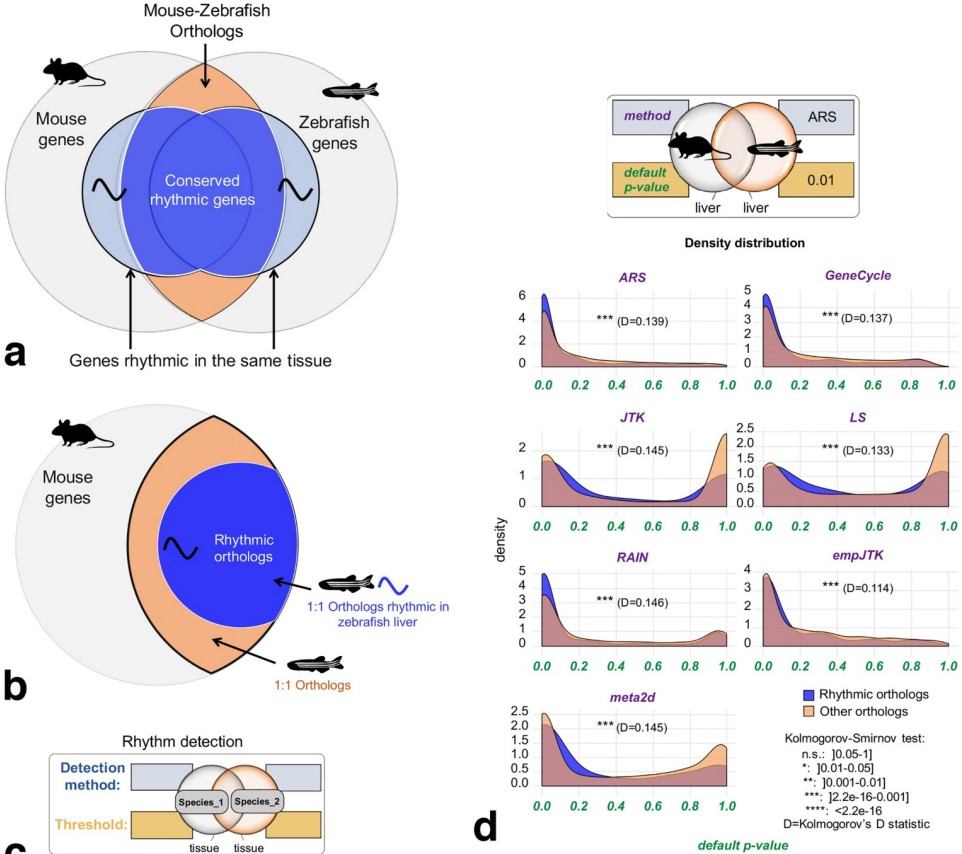

**Fig 5. Signal of evolutionary conservation of rhythmic gene expression.** Orthologous genes detected as rhythmic in the same organ of two species have a stronger statistical signal of rhythmicity than those not detected as rhythmic in at least one species. **a)** Mouse and zebrafish share orthologous genes, some of which are rhythmic in the homologous tissues. **b)** Method used for ortholog benchmarking, as in panel **d**: From all mouse genes, only mouse-zebrafish one-to-one orthologs are kept. Considering the liver, these orthologs are separated into two groups: genes for which the ortholog is detected as rhythmic in zebrafish liver, called rhythmic orthologs; and the remaining one-to-one orthologs. **c)** Chart providing the legends to inform about the method and the threshold used to call genes rhythmic for each condition (species and tissue). **d)** *p*-values density distribution of rhythmic orthologs vs non-rhythmic orthologs obtained for the seven methods applied to mouse liver data. Mouse-zebrafish orthologs, that are detected rhythmic in zebrafish liver, are significantly more enriched in small *p*-values in mouse liver, for all methods (Kolmogorov-Smirnov test *p*-values < 0.001). Images credit: Anthony Caravaggi (mouse), Ian Quigley (zebrafish).

be conserved between orthologs, but rather that true rhythmic genes should have more rhythmic orthologs than false-positive predictions. For a given *p*-value threshold, each method detects a certain number of rhythmic genes (genes with *p*-value under the threshold). At each threshold we calculated the proportion of orthologs rhythmic in species2 among one-to-one species1-species2 orthologs, as defined in Fig 6a. This proportion allows to assess how each method is able to detect the conservation of rhythmicity and can be calculated for each *p*-value threshold. The benchmark set is composed of orthologs detected rhythmic in the second species, called rhythmic orthologs. To define this set, we chose a rhythmicity detection method among ARS, empJTK, and GeneCycle, in agreement with results of previous sections, and a *p*-value threshold of 0.01.

A risk is that orthologs have conservation of gene expression levels and that there is a bias towards calling highly expressed genes "rhythmic". To control for this in the benchmarking, we added a "Naive" method based only on expression levels. This Naive method simply orders

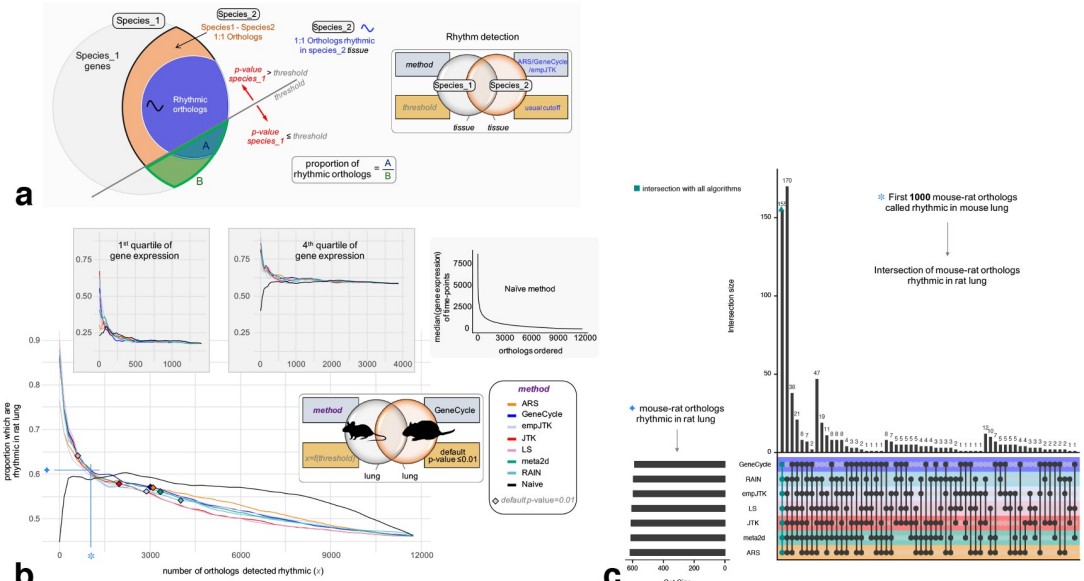

**Fig 6. Only strong rhythmic signals of gene expression are relevant.** Methods designed for rhythm detection in gene expression show an advantage only for the genes with a strong rhythmic signal, i.e. related to very small *p*-values. For a fixed number of top genes called rhythmic, all the methods, despite their design differences, retrieve approximately the same proportion of biologically functional rhythmic genes and the same genes themselves. **a)** Method to obtain figure b: For a given *p*-value threshold, each method detects a certain number of rhythmic genes (genes with *p*-value ≤ threshold). At each threshold, we calculate the proportion of orthologs rhythmic in species2 (A) among one-to-one species1-species2 orthologs (B). The benchmark set is composed of one-to-one orthologs detected rhythmic in the second species (using method ARS, GeneCycle, or empJTK), called rhythmic orthologs. **b)** Variation of the proportion rhythmic orthologs/all orthologs in mouse as a function of the number of mouse orthologs detected rhythmic, for each method applied to the mouse lung dataset. The benchmark gene set is composed of mouse-rat orthologs, detected rhythmic in rat lung by the GeneCycle method with default *p*-value ≤ 0.01. The black line is the Naive method which orders genes according to their median expression levels (median of time-points), from highest expressed to lowest expressed gene, then, for each gene, calculates the proportion of rhythmic orthologs among those with higher expression. The proportion of the benchmark set among one-to-one orthologs is higher for highly expressed genes (4th quartile) than for lowly expressed genes (1st quartile) (∼ 60% vs ∼ 20% respectively). Diamonds correspond to a *p*-value threshold of 0.01. **c)** Upset diagram showing the number of rhythmic orthologs (figure **a**) called in common by the methods among the first 1000 mouse-rat orthologs that are called rhythmic in mouse lung. Images credit: Anthony Caravaggi (mouse) and Public Domain for other images (from PhyloPic).

genes (orthologs here) according to their median expression levels (median of time-points), from highest expressed to lowest expressed gene, then, for each gene, we calculated the proportion of rhythmic orthologs among those with higher expression. We also present results for subsets obtained from the division in four quartiles of expression levels. Fig 6b shows the variation of the proportion defined above as a function of the number of orthologs detected rhythmic, obtained for each method applied to the mouse lung dataset. The benchmark gene set was defined by mouse-rat orthologs, detected rhythmic in rat lung by the GeneCycle method (default *p*-value ≤ 0.01). Genes are given by order of their detection by the methods. The genes with small *p*-values, i.e. with a strong signal of rhythmicity, had a high proportion of rhythmic orthologs. Importantly, for all methods, this proportion was higher than that obtained from the Naive method (Fig 6b). Results are consistent in almost all species comparisons, with exceptions for cerebral tissues (S4 File). However, the thresholds of 0.01 are to the right of the intersection between the curves of rhythm detection methods and the Naive, except for LS. This means that, for an apparently reasonable threshold (*p*-value ≤ 0.01), ranking genes by expression level performed "better" than all methods designed specially for rhythm detection. We made the same observation using an FDR-based threshold (FDR≤ 0.01 or FDR≤ 0.1, S1 File). These results

imply that even with a stringent *p*-value or FDR threshold, such as 0.01, the rhythmic nature of some of the genes considered rhythmic is not relevant. These rhythm detection methods were relevant only for genes with very high signal of rhythmicity, where they performed better than a Naive method. Finally, for the top 1000 mouse-rat orthologs detected as rhythmic in mouse lung, all the methods reported a similar proportion of rhythmic orthologs, around 62%, mainly highly expressed genes (fourth quartile of gene expression) (Fig 6b). And the overlap between these orthologs was largely detected by all methods (Fig 6c). Thus, for genes with a high signal of rhythmicity, all methods performed similarly to detect the tissue-specific conservation of gene expression rhythmicity. Similar results were obtained for other species comparisons (S5 File).

## Discussion

The methods designed for rhythm detection in gene expression perform similarly and only for strong rhythmic signal. In this study, we show that orthologous genes detected as rhythmic in the same organ of two species have a stronger statistical signal of rhythmicity than those detected as not-rhythmic in at least one species. These results support our hypothesis that the nycthemeral rhythmicity at the gene expression level is biologically functional, and that this functionality is more conserved between orthologous genes than between random genes. We define the nycthemeral transcriptome as all genes displaying a rhythmic expression repeated every 24 hours. In order to assess the performance of seven methods to detect these rhythms, we used this concept of conservation of the rhythmicity between species for benchmarking. We employed genes whose orthologs had a rhythmic expression called in the same homologous organ as a proxy for a true positive set, as done in some previous benchmarks. For instance, Rosikiewicz et al. [36] assessed the quality of microarrays quality control methods based on evolutionary conservation of expression profiles, and Kryuchkova et al. [37] benchmarked tissue-specificity methods in the same way. This approach based on real data, also used by Boyle et al. [3] to solve the issue of weak overlap between the same tissues from the same species from different experiments, avoids relying on simulations which tend to favor methods using the same model, e.g. the same patterns, and has the advantage of not being based on specific assumptions, other than general evolutionary conservation of function. By taking into account the tissue-specificity of rhythmic gene expression and different species comparisons, we show that no method is strongly favoured. For instance, one would have expected that the added features of RAIN and empJTK allowing then to detect more diverse patterns than a classical sinusoidal would have favored them. But this flexibility did not provide them any advantage in the benchmark. Furthermore, the comparison of the methods with a 'Naive' one, uninformed about rhythmicity, shows an advantage for informed methods only for the genes with a strong rhythmic signal. Thus, only genes with a strong rhythmic signal, i.e. the top genes called rhythmic, can be considered as relevant. Even if the threshold of "relevance" of these genes is dependant on the evolutionary distance of the species compared, these results suggest a call for caution about the results of previous studies reporting or based on large sets of rhythmic genes. For the same number of genes called rhythmic, all the methods, despite their design differences, retrieved approximately the same proportion of biologically functional rhythmic genes (Fig 6b) and the same genes themselves (Fig 6c).

### The issue of significance

For the same *p*-value threshold, the number of genes called rhythmic is different from one method to another, with a large proportion of these genes detected rhythmic by only one or a few methods. But, if we consider the top genes called rhythmic for each method, without

taking into account any *p*-value threshold, the overlap of rhythmic genes become strong between the methods (Fig 4c and 4d). This highlights an issue with the meaning of the *p*-value and the associated thresholds used. This is directly related to the issue of correction that needs to be improved in this field. When a smaller number of top rhythmic genes is used, the overlap between methods becomes weaker (Fig 4c and 4d). Thus, the order of calling genes rhythmic is different from one method to another. Finally, since methods performed better than a Naive method only for genes with a strong rhythmic signal, we can not conclude for the relevance of the other genes called rhythmic, even when they have very low nominal *p*-values.

### ARS, empJTK, and GeneCycle produce consistent *p*-values

ARS, empJTK, and GeneCycle were the methods that showed the best behavior on real and randomized data (single species tests). They were the only methods displaying both a uniform distribution of their *p*-values under the null hypothesis, and a left-skewed distribution when applied to real data. For empJTK, its default correction allowed to produce these expected results. However, each of these three methods is conceptually completely different, which indicates that there is not one conceptual framework which dominates rhythmic gene detection. ARS combines time-domain and frequency-domain analyses. GeneCycle, which is the robust spectrum function of the R package, is based on a robust spectral estimator which is incorporated into the hypothesis testing framework using a so-called g-statistic together with correction for multiple testing. And, empJTK improves the original JTK including additional reference waveforms in its rhythm detection. The other methods all presented major issues. LS has a right-skewed distribution of its initial uncorrected *p*-values suggesting an invalid null hypothesis. JTK, RAIN, and meta2d had also issues with their null hypothesis displaying left-skewed distributions of their uncorrected *p*-values. Their default adjustment was excessive, favoring high *p*-values obtained after correction. Hutchison and Dinner [32] observed this on simulated data, and proposed that it was due to non independence of measurements from the same time series.

### Biological insight into gene rhythmicity in animal tissues

Our results support the hypothesis that rhythmic genes are largely enriched in highly expressed genes (Table 2). Experimental noise that would mask the rhythmic signal of lowly expressed genes could also explain this result in part, especially considering that the datasets with good sampling used microarray technology. BooteJTK compares the noise to the amplitude of a time series, in addition to evaluating the rank order of the values, and thus might provide a more relevant rhythm detection by improving the variance estimation from biological replicates [38]. The observation of known cycling genes in different organs seems to indicate different profiles of rhythmicity possible for the same gene. For instance, Npas2 displays a cosine shape in kidney and lung, and a peak/box shape in liver and muscle (Fig 2a). This observation suggests that methods might perform differently depending on the organ studied. This is also one of the reasons why all our analyses were made for homologous organs.

   In mouse-baboon comparisons, there were no significant differences of *p*-value density between rhythmic and non-rhythmic orthologs in cerebral tissues: brain stem, cerebellum, and supra-chiasmatic nucleus, except for the hypothalamus (S4 File). This could be explained by the fact that there are only low amplitudes of expression of clock genes and few rhythmic genes in almost all brain regions. This is assumed to be due to an inefficient synchronization of individual cellular oscillators in brain cells to avoid noise into the synchronizator element [39]. In addition, it could also be an essential aspect for intrinsic brain processes which could require a constant expression of most genes.

**Table 2. t-test comparing the expression levels between rhythmic ($p$-value $\leq 0.005$) and non-rhythmic genes (randomly chosen same number of genes among those with $p$-value $> 0.01$), in mouse liver dataset (microarray).**

| Method | group | n | Mean |
|---|---|---|---|
| ARS | rhythmic | 4019 | 1151.2 |
| | non-rhythmic | 4019 | 383.6 |
| **t-test** t = 25.4 $p$ <2.2e−16 df = 7021.4 | | | |
| meta2d | rhythmic | 4520 | 1113.2 |
| | non-rhythmic | 4520 | 398.5 |
| **t-test** t = 24.8 $p$ <2.2e−16 df = 8050.1 | | | |
| empJTK | rhythmic | 3373 | 1113.9 |
| | non-rhythmic | 3373 | 442.5 |
| **t-test** t = 20.1 $p$ <2.2e−16 df = 6260.8 | | | |
| RAIN | rhythmic | 5044 | 1066.3 |
| | non-rhythmic | 5044 | 384.2 |
| **t-test** t = 25.4 $p$ <2.2e−16 df = 8935.3 | | | |
| JTK | rhythmic | 2646 | 1214.4 |
| | non-rhythmic | 2646 | 454.3 |
| **t-test** t = 19.6 $p$ <2.2e−16 df = 4742.9 | | | |
| LS | rhythmic | 736 | 1500.3 |
| | non-rhythmic | 736 | 526.5 |
| **t-test** t = 12.6 $p$ <2.2e−16 df = 1351.2 | | | |
| GeneCycle | rhythmic | 4145 | 1082.8 |
| | non-rhythmic | 4145 | 425.6 |
| **t-test** t = 22.0 $p$ <2.2e−16 df = 7622.8 | | | |

## The importance of having an informative dataset

Because of the cost and complexity of circadian experiments, time-series datasets of gene expression in animals are rare, especially in the same experimental conditions. Algorithms must be able to deal with little data, but importantly experiments should take into account the algorithms' sensitivity. All algorithms appeared to produce relatively poor $p$-values distributions when applied to the available Drosophila or baboon datasets, and, for the baboon dataset, were almost always less efficient than the Naive method (S1 File). This baboon dataset is probably not very informative, which raises questions about the biological conclusions from the associated study [1]. With only one replicate per time-point, over only one cycle of 24 hours, the algorithms are unable to detect repetitive patterns. Variations over a single 24 hours cycle appear to be insufficient to detect rhythmic signal, when there is no evidence of repetition over several cycles. Moreover, each data comes from different outbred individuals. The variations of gene expression between two time-points can be due to individual variations or real oscillation within the population. It is possible that sinusoidal patterns with a continuous trend over successive time-points could be detected without replicates, although power will be lacking, but patterns such as the peak pattern will be extremely sensitive to inter-individual variation. Fig 3 generally suggests that datasets with one replicate per time-point over a unique cycle of 24 hours do not provide enough information that would allow to correctly detect the rhythmicity. It seems that ARS in peculiar is very sensitive to the repetitive nature of profiles. Of note, for time-series with low sampling frequency, a recent improvement of empJTK, called BooteJTK, allows to detect rhythms robustly relative to sampling frequency [38]. Thus, if only one 24h cycle is feasible, several biological replicates must be favored. Our results support the conclusions of Hutchison et al. [6] who indicate that for a fixed number of samples, better sensitivity and specificity are achieved with higher numbers of replicates than with higher

sampling density. We propose that future experiments should produce data with two biological replicates per time-points as a strict minimum. Obviously, we suggest considering biological replicates as new cycles within one replicate, as proposed in recent guidelines [9]. GeneCycle, and to a lesser extent empJTK, were the most robust methods when applied to weakly informative datasets. Thus, the performance of the algorithms is dependent on techniques and experimental designs used for the experiments. The optimization of experimental plans (see section Recommendations) could improve the methods' performance for the detection of rhythmically expressed genes. Moreover, we recommend producing data over at least two cycles to be sure of the repetitive nature of profiles, and to avoid a potential random influence of the shared environment, which might be considered rhythmic since it affects all replicates. Finally, contrary to the mouse experiment, the rat experiment has been done under zeitgeber conditions which have most likely resulted in more genes being expressed rhythmically, so in proportion, more periodic patterns. This might explain the higher density of small $p$-values obtained for the rat dataset (Fig 3a). Comparison between these two datasets is not expected to have removed the signal, since we found a good correlation of $p$-values obtained between two conditions, light-dark versus dark-dark, in data produced from the same experiment (S1 File).

## Limitations and improvement of methods

ARS and GeneCycle need complete chronological data and cannot deal with biological replicates. Except for LS, RAIN, and empJTK, all other methods studied here assume equally spaced time-points. Furthermore, ARS needs an integer sampling interval with regular time-series datasets and cannot deal with missing values, or with several replicates per time-point. In this study, ARS appeared to be efficient only for the dataset with at least two cycles of data. Indeed it produced aberrant $p$-value distributions when applied to datasets restricted to one cycle of 24 hours. But, for these datasets, all algorithms behaved poorly. The improvement of JTK by empJTK produced much better results than the original JTK algorithm. It is possible that the improvement of RAIN suggested by Hutchison and Dinner [32], which allows to produce uniform $p$-values distribution under the null, might similarly improve the results of RAIN. We believe that LS could be a very interesting method if its null hypothesis could be clarified and would thus provide $p$-values with proper behavior. LS has advantages that other algorithms don't. For instance, it can deal with irregular intervals, missing data, and has been shown to stay efficient on small sample size [27], which constitutes one of the big issues of circadian transcriptomic data. On the other hand, relative to JTK, ARS, or MICOP methods, LS has also been shown to be highly sensitive to the increase of sampling intervals and to noise for proteomic data [40].

A good method must, at least, display a uniform distribution under the null hypothesis, and a classic skewed distribution when applied to full dataset or even more to known cycling genes. It should also be able to detect efficiently rhythmic orthologs, which represent an important part of the functionally relevant nycthemeral rhythmicity. In this study, we did not assess the amplitudes, phases, and precise period provided by the algorithms. We only analysed the performance of methods for nycthemeral or circadian rhythms in gene expression data, and cannot conclude directly for ultradian or seasonal rhythms, and for other types of datasets which are not gene expression data.

## Recommendations

**Experimental design.**   1.  Always use at least 2 biological replicates per time-point.

2.  One full period sampled is the minimum required. Two periods are to be preferred.

3. Favor time-points number (small temporal resolution) over transcriptome profiling quality (e.g., microarray vs RNAseq).

4. Favor regular sampling because only few algorithms can deal with irregular interval time-series.

5. For a fixed number of samples, favor higher numbers of replicates over higher sampling density (see also [6]).

**Recommendations about the choice of rhythm detection method, the arrangement of the time-series dataset, and the interpretation of results based on these seven methods studied.**

1. Only genes with a strong rhythmic signal should be considered as relevant. By "strong" we mean the top genes called rhythmic, knowing that the threshold of $p$-value $\leq 0.01$ is already not stringent enough for some methods.

2. Take into account that detected rhythmic genes are strongly enriched in highly expressed genes.

3. LS could be a good candidate to improve.

4. Favor ARS, GeneCycle, or empJTK with default parameters.

5. Consider biological replicates as new cycles with one replicate.

6. Check by eye for rhythms of known circadian genes.

7. Never duplicate and concatenate data before running algorithms [9].

8. Never consider technical replicates as biological replicates [9].

## Methods

### Pre-processing

For each time-series dataset, only protein coding genes were kept. For microarrays, we removed probIDs which were assigned to several GeneIDs. ProbIDs or genes which contained one or several missing values have been removed, allowing comparison between all methods even those which can not deal with missing values. Genes with no expression (= 0) at all time-points were also removed. For each species dataset, we only kept comparable conditions to other species of reference. Tissues separated in sub-tissues such as adrenal gland in adrenal cortex and adrenal medulla in baboon experiment were removed.

For each condition (species and tissue), several datasets have been built: i. the full original dataset; ii. the first and fourth quartiles of the median gene expression level of the original data; iii. randomized data (time-points redistributed randomly); iv. randomized data restricted to the first and fourth quartiles of the median gene expression level; and v. a subset of known cycling genes when such data was available (8 to 99 genes according to species).

### Normalization by Z-score

The normalization of gene expression values by Z-score transforms the pre-processed data such that for gene $i$ with the original expression value at time-point $j$ is *gene.ij*, we have:

$$gene.ij.normalized = gene.ij - xi$$

with $xi = mi - \frac{Zi}{j}$. $mi$ is the mean expression of gene $i$: $mi = \frac{\sum^{gene.ij}}{j}$; and $Zi = \frac{mi-m}{sd}$; $m$ and $sd$ being the mean and the standard deviation of the original full dataset.

## Orthology relationships

For each species comparison, orthologs relationships have been downloaded from OMA [41]. For simplicity, we only considered one-to-one orthologs. In species comparisons, we only kept orthologous genes that had available data in both species.

## Algorithms and packages

`MetaCycle` R package was performed with parameters: minper = 20h and maxper = 28h. This package incorporates the 3 algorithms to detect rhythmic signals from time-series data-sets: ARSER (ARS), JTK_CYCLE (JTK), and Lomb-Scargle (LS). It also provides meta2d that integrates analysis results from multiple methods based on an implementation strategy (see "Introduction to implementation steps of MetaCycle" in MetaCycle documentation for more details). ARS does not deal with several replicates per time-point. To not introduce biases, we only kept one replicate for ARS performing when the dataset was provided with several replicates per time-point.

`Rain` R package was performed with parameters: period = 24h, period.delta = 4h (width of period interval), and method = 'independent'. In order to obtain unadjusted pvalues as output, we modified the source code of the rain and MetaCycle R packages.

`Empirical-JTK` (empJTK) was executed by running bash commands with parameters: cosine waveform, 24 hours' period, look for phases every 2 hour from 0 to 22 hours and look for asymmetries every 2 hour from 2 to 22 hours (GitHub alanlhutchison/empirical-JTK_CYCLE-with-asymmetry). It is important to run empJTK with python version 2.7.11. Raw $p$-value correspond to `P` output (P-value corresponding to Tau, uncorrected for multiple hypothesis testing), and default $p$-value correspond to `empP` output (min($p$-value calculated from empirical null distribution, Bonferroni)).

`GeneCycle` R package [22] was downloaded from CRAN. We used the robust.spectrum function developped by [21] that computes a robust rank-based estimate of the periodogram/correlogram.

Plots have been created using `ggplot2` R package (version 3.1.0); Upset diagrams using `UpSetR` R package (version 1.3.3) [42]; and Venn diagram using `venn` R package (version 1.7).

## Statistical analysis of rhythmic gene expression

All the rhythm detection methods (See Materials) were applied to each pre-processed dataset, producing a list of $p$-values as output. Then, for each gene having several results (ProbIDs or transcripts), we combined $p$-values by Brown's method using the EmpiricalBrownsMethod R package. Thus, for each dataset, we obtained a unique $p$-value per gene. Whenever the per-gene normalization was not necessary (unique data for all genes), we obtained the original $p$-value for each gene. FDR is the false discovery rate adjustment of default $p$-values using `p.adjust` R function.

## Naive method

The Naive method is only based on expression levels of genes and is not informed about rhythm detection. It simply orders genes according to their median expression levels (median of time-points), from highest expressed to lowest expressed gene. Then, for each gene $i$, we

calculate the proportion of rhythmic orthologs among those with higher expression, i.e. among the genes from the highest expressed one to the gene *i*.

**Availability of data and scripts.** The data and scripts for reproducing plots and analysis are available at https://github.com/laloumdav/rhythm_detection_benchmark.

## Materials

### Ethics statement

We had ethical issues to use olive baboon data since these data needed the sacrifice of twelve baboons. We would like to remind that such data would have been impossible to get in Switzerland where the primate research is prohibited. We still support Switzerland ethical considerations in matter of animal research and think that the scientific knowledge can not justify an irresponsible employment of life on earth. While being aware that our results would have been less robust without these data and that these considerations on primate could also be generalized to other living organisms.

### Datasets

*Mus musculus* **(13 tissues).** Raw microarray and RNA-seq data, from [2], was downloaded from GEO accession (GSE54652). Microarray gc-rma normalized data was sent by Katharina Hayer from CircaDB database [43]. Expression values were already normalized between biological replicates to average out both biological variance between individual animals and technical variance between individual dissections. RNA-seq data was already normalized using DESeq2. Data was obtained for adrenal gland, aorta, brain stem, brown adipose, cerebellum, heart, hypothalamus, kidney, liver, lung, muscle, SCN (only microarray), and white adipose. Probesets on the Affymetrix MoGene-1.0-ST-V1 array were cross-referenced to best-matching gene symbols by using Ensembl BioMart software.

*Papio anubis* **[olive baboon] (11 tissues used).** RNA-seq data from [1] was downloaded already normalized by using DESeq2. Read counts per gene were calculated using Feature-Counts. We kept data for aorta, brain stem, cerebellum, heart, hypothalamus, kidney, liver, lung, muscle, SCN, and white adipose tissues. Data were already provided with Ensembl gene symbols.

*Rattus norvegicus* **(lung).** Raw microarray data from [44] was downloaded from GEO accession (GSE25612). Over 3 days, 54 samples were extracted in light-dark condition with a temporal resolution closer for some time-points (See paper for more details). Contrary to the study, we still considered the 3 successive days samples as successive days measurements. ARS, JTK and RAIN methods don't operate with irregular time-series. We normalized time-series by calculating the mean value of irregular time-points to obtain regular time-series. rma normalization was performed using the rma R-package. Probesets on the Affymetrix 230-2-probe array were cross-referenced to best-matching gene symbols by using Ensembl BioMart software.

*Dano rerio* **(liver).** Raw microarray data from [3] was downloaded from GEO accession (GSE87659). Data was already rma-normalized, averaged gene-level signal intensity, and already cross-referenced to best-matching transcript symbols.

*Anopheles gambiae* **(head and body).** Raw microarray data from [33] was downloaded from GEO accession (GSE22585). Non-blood fed female mosquito heads and bodies were collected under light dark and constant dark conditions. We only used data collected in LD condition, except for the comparison of both conditions (LD versus DD). We normalized data using the rma R package and cross-referenced to best-matching gene symbols by using VectorBase software.

*Aedes aegypti* **(head).**   Raw microarray data from Ľeming was downloaded from GEO accession (GSE60496). Non-blood fed female mosquito heads were collected under light dark and constant dark conditions. We only used data collected in LD condition. NimbleGen Aedes aegypti 12plex array already rma normalized were provided with VectorBase geneIDs.

*Drosophila melanogaster* **(head and body).**   RNA-seq data from [45] was downloaded from GEO accession (GSE64108). They measured RNA concentrations in the head and body of 3-, 5-, and 7-week-old adult flies in ad libidum feeding or 12-hour time-restricted feeding conditions. We only used data from ad libidum feeding condition of 5-week-old adult flies with best temporal resolution.

## Cross-referenced gene IDs and known cycling genes

GeneID, protein coding status, ProbSetID, transcriptsID were downloaded from Ensembl [46] or VectorBase [47] using BioMart.

Known cycling genes were obtained from the KEGG [48] or FlyBase [49] database:

- KEGG circadian entrainment entry pathway for the mouse (mmu04713) and the rat (rno04713). This is the pathway by which light activates SCN neurons and the resulting signaling cascade that leads to a phase resetting of the circadian rhythm generated in these neurons. Most of these genes are not involved in generating the rhythm itself and as such cannot be called 'clock genes'.

- KEGG circadian rhythm entry pathway for the baboon (human hsa04710), and Anopheles (aga04711)

- FlyBase circadian rhythm entry pathway for Drosophila (GO:0007623).

## Supporting information

**S1 Table. Gene expression time-series datasets.** Gene expression time-series datasets that come from circadian experiments. We kept data from healthy, wild-type individuals for these seven species, allowing comparisons among vertebrates and among insects. We preferred data from light-dark (LD) and ad-libitum experimental conditions whenever possible as providing a better representation of wild conditions. LD for regular alternation of light and darkness each 24h; and DD for continuous darkness usually after an entrainment to a 12h:12h light: dark.
(XLSX)

**S1 File. Supplementary results.**
(PDF)

**S2 File. Density distribution of raw and default *p*-values obtained for the seven rhythm detection methods applied to vertebrate datasets.** Density distribution of *p*-values obtained before (raw) and after the default correction (software) for the seven methods applied to each vertebrate dataset, sub-categorized in: i. randomized data which represents the null hypothesis; ii. randomized data restricted to the first and fourth quartiles of the median gene expression level, to check for the impact of expression level under the null; iii. the full original dataset; iv. the first and fourth quartiles of the median gene expression level of the original data; and v. a subset of known cycling genes when such data was available (8 to 99 genes according to species). The default *p*-values of ARS, GeneCycle, and LS are uncorrected.
(PDF)

**S3 File. Following S2 File.**
(PDF)

**S4 File. Signal of evolutionary conservation of rhythmic gene expression in vertebrates.** *p*-values density distribution of rhythmic orthologs vs non-rhythmic orthologs obtained for the seven methods applied to different vertebrate datasets. Orthologous genes detected as rhythmic in the same organ of two species have a stronger statistical signal of rhythmicity than those detected as not-rhythmic in at least one species. From all species_1 genes, only species_1-species_2 one-to-one orthologs are kept. Considering homologous tissues, these orthologs are separated into two groups: genes for which the ortholog is detected as rhythmic in this tissue of species_2, called rhythmic orthologs; and the remaining one-to-one orthologs.
(PDF)

**S5 File. Variation of the proportion A/B as a function of the number of orthologs detected rhythmic, obtained for each method applied to different vertebrate datasets.** The benchmark gene set is composed of species_1-species_2 orthologs, detected rhythmic in the homologous tissue of species_2 by the ARS, GeneCycle, or empJTK method with default *p*-value $\leq 0.01$ or 0.05. See Fig 6 for definitions of sets A and B. The black line is the Naive method which orders genes according to their median expression levels (median of time-points), from highest expressed to lowest expressed gene, then, for each gene, calculates the proportion of rhythmic orthologs among those with higher expression.
(PDF)

**S6 File. Results in insects.**
(PDF)

## Acknowledgments

We thank Paul Franken for useful discussions, as well as all members of the Robinson-Rechavi lab.

## Author Contributions

**Conceptualization:** David Laloum, Marc Robinson-Rechavi.

**Data curation:** David Laloum.

**Formal analysis:** David Laloum.

**Funding acquisition:** Marc Robinson-Rechavi.

**Methodology:** David Laloum, Marc Robinson-Rechavi.

**Writing – original draft:** David Laloum.

**Writing – review & editing:** Marc Robinson-Rechavi.

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
