## [Decision Letter · Decision Letter 0]

7 Oct 2019

Dear Dr Robinson-Rechavi,

Thank you very much for submitting your manuscript 'Methods detecting rhythmic gene expression are only reliable for strong signal' for review by PLOS Computational Biology. Your manuscript has been fully evaluated by the PLOS Computational Biology editorial team and in this case also by independent peer reviewers. The reviewers appreciated the attention to an important problem, but raised some substantial concerns about the manuscript as it currently stands. While your manuscript cannot be accepted in its present form, we are willing to consider a revised version in which the issues raised by the reviewers have been adequately addressed. We cannot, of course, promise publication at that time.

Sincerely,

Attila Csikász-Nagy

Associate Editor

PLOS Computational Biology

Bjoern Peters

Benchmarking Editor

PLOS Computational Biology

[LINK]

Reviewer's Responses to Questions

**Comments to the Authors:**

Reviewer #1: Comments for PCOMPBIOL-D-19-01419

————————

This manuscript works to compare different rhythm detection methods using identification of orthologous rhythmic genes between species as a measurement of method accuracy. It identifies methods that perform better at this, and finds an association with expression level. It also re-enforces the previous findings that several of the methods do not have well-behaving p-values.

This thorough and extensive manuscript will be a beneficial addition to the circadian rhythm detection literature but would be significantly improved by addressing the major and minor issues which are within the scope of the paper, detailed below:

The manuscript can be primarily improved along these points

1) Clarification of the definition of strong signal

If I am not mistaken, the way you are defining “strong rhythmic signal” is just as genes with high expression. Why are you doing it this way and not using amplitudes? I can imagine a time series with high expression but small amplitude (1000 -> 1002 -> 1000 -> 998 -> 1000), and there are several papers with suggestions on how to define amplitude (Max - min, SD, Sqrt(2)*(Max-min) etc. I think this ends up making your argument a little misleading. If you mean “high expression genes” then say that instead of “strong rhythmic signal”.

Building on this, it is not clear to me why we should be concerned that rhythm detection methods are preferential to high-amplitude time series (that is, time series with a high signal(amplitude)-to-noise ratio. If anything, this seems like it would be preferable, and in fact methods like Bootstrap eJTK from Hutchison et al 2018 try to explicitly incorporate this.

2) Clarification of the definition of reliable

Though it is in your title, you do not actually define what you mean by “reliable”. Past definitions have used “able to distinguish signal from noise” as a proxy, or “able to identify the same gene when downsampled from the same biological experiment (Hutchison et al 2018) but here it seems you use “find genes rhythmic in one organism to have rhythmic orthologues in a different organism”. It is not clear that this gives more information about the differences or limitations of the methods or biological insight into differences between the organisms, tissues, or experimental conditions being compared. I think you need to make a stronger argument why we should expect the particular genes that you are comparing across organisms should be rhythmic, and the failure to identify them as such is a limitation of the methods as opposed to a interesting biological finding. As mentioned above, comparing across several organisms at once might be a way to strengthen this argument, or looking at “known rhythmic genes” such as the KEGG annotated ones (though see below for concerns about using these genes with the expectation that their expression profiles will be rhythmic).

3) Comparison across species only and not experimental design

While I think your use of orthologous genes in homologous tissues is an useful one, I think it is less useful when you confound the results comparing experiments done in LD vs. DD. While there should be an overlap for LD vs DD, the literature has already shown sufficient difference that making the comparison you have chosen with mouse DD vs rat LD weakens your results and your argument. I think your use of orthologous genes would be stronger if you used homologous organs in two different species where both organisms are tested under LD or DD conditions.

If you are going to compare across experimental designs and species simultaneously, then it might be interesting to look across more than two species at a time, as using that overlap among several species should mitigate some of the confounding from looking across experimental designs and be a stronger criterion on which to compare different methods.

4) Use and interpretation of p-values

I think something else to note here is that the expected proportion of overlap for these methods run on the same data is already between 60-70% (Hutchison et al. 2018), even within the same tissue, lab, and experimental collection. In your Figure 7, though, you show quite high levels of overlap. This discrepancy may be explained because you are not multiple-hypothesis correcting your rhythmicity results. This brings up a concern of its own. In lines 448-450 you mention that 0.01 might not be stringent enough of a cutoff. Why aren’t you multiple-hypothesis correcting in that case? Do your results change when you multiple-hypothesis correct?

Building on this, in your section at line 315 “The issue of significance” and “ARS, empJTK, and GeneCycle produce consistent p-values”, you correctly distinguish between the methods that generate correct p-values and those that don’t, but I think it might be worth discussing further the meaning and point of p-values as heuristics for rhythmicity. At a small scale, the overlap of the top 100 genes might vary widely, but be important because it will determine what individual genes to next conduct an experiment on, be it a knock out, or a new chronotherapy target. At that level, differences between the methods become important, and whether the gene has a 5% of being a false positive or 0.5% chance becomes very important. At a large scale, it might determine what biological processes to experimentally intervene upon, and the false positive rate becomes important there as well. I think merely ranking the order of genes misses this aspect of the rhythm detection methods.

5) Alternative conclusions of your results

The title of your paper is “Methods detecting rhythmic gene expression are only reliable for strong signal”. I do not know if this statement can be distinguished from other statements that could be drawn from the same data, such as

- For genes with low expression, it is difficult to tell if their rhythmicity is conserved across species

- Rhythmicity is not conserved among genes with low expression.

- If orthologous genes have discordant expression levels, it is less likely that they will both be identified as rhythmic

- Methods detecting rhythmic gene expression agree more often for highly expressed genes

————————

Minor points:

Figure 1a:

I do not understand the placement of “Positive elements” as it seems to me it is next to an arrow that is a repressor

Figure 1b:

i) I don’t know what the goal of reporting the raw p-values is for JTK, meta2d, RAIN, and empJTK. It is clear from the design of the methods that they will all under-estimate p-values before correction, which you show, but I do not know what value to your argument or the discussion is added by having the raw p-values included in what is already a very busy (but very informative) figure.

ii) I think it would be worth including a subset image on the far right with a high right peak, to demonstrate the case where false negatives predominate and signal is discarded. It is arguably as important for a method to have a low false negative rate as a low false positive rate, and there are several examples in the figure (meta2d, JTK, RAIN) that show large right-sided peaks.

iii) Do we expect the RNA expression of all the ‘KEGG circadian entrainment’ genes to have rhythmic RNA expression? There are several cases where the protein levels are rhythmic and the RNA is not, cases where rhythmic protein modifications drive circadian involvement in genes that do not have RNA expression rhythmicity, and cases where protein levels are constant but the protein binds with a rhythmically expressed protein. All these cases would be indications for inclusion in in ‘KEGG circadian

Line 119:

You argue here that our analyses should not be biased towards high expressing time series, but is there i) a biological argument that these are more relevant or ii) an analytical argument that distinguishing strong signals that rise above the noise is an important thing to look for (the idea behind BooteJTK Hutchison et al 2018)?

Figure 2a

i) The figure might be legible if instead of p-values you plotted -log10(p-values). If you’re trying to plot a uniform distribution it makes sense to deal with p-values in their original state, but for these scatterplots I think it would help delineate signal from noise.

Figure 3

The caption should read “Fewer time points per cycle” instead of “Less time points”

Figure 5c and 5d

After some time looking at the figures, I still don’t understand how if the “upset diagrams” are showing the intersection of genes in the top 6000 across methods, as you reduce methods I would expect you have to have higher numbers of intersection (more genes would be rhythmic in both empJTK and ARS than in empJTK, ARS, and LS, since some genes rhythmic in empJTK and ARS won’t be rhythmic in LS, and therefore excluded from that intersection). That I have put in a good effort to understand these plots and still cannot suggests that the explanation should be more explicit, or the data should be replotted.

Figure 6d

Is this distribution comparison the best way of comparing across tissues? I would expect a scatterplot comparing p-values for each orthologous gene would better support your argument for overlap or divergence of methods.

294-295: Give X et al before citation in the same way you do in 297

412: This statement is untrue. empJTK accepts as inputs any time points as the header.

326: Are you saying we shouldn’t pay attention to rhythmic genes with low expression? Are there any biologically plausible reasons that genes that are not rhythmic across species would have high expression vs. low expression levels?

Reviewer #2: Review – Methods detecting rhythmic gene expression are only reliable for strong signal.

In this manuscript the authors focus on the importance of identifying correctly the nycthemeral transcriptome or the rhythmic expression patterns of certain genes, which has been linked with relevant biological processes. The authors enumerate several methods commonly used to detect rhythmic genes and aim at benchmarking these different approaches (7 different methods) by evaluating their performance in real datasets compared to randomized ones and also by including homology context (evolutionary conservation). The study objective is twofold: define an approach to benchmark future methods based on biological data rather than simulated data, and provide recommendations on the design of time-series experiments as well as the choice of detection method.

The manuscript is nicely written, clear and detailed. The stated problem is relevant and clearly demonstrated in the statistical evaluation of the results (p-values distributions analysis) from the selected methods (JTK, LS, ARS, meta2d, empJTK, RAIN and GeneCycle), which indeed show different ability to detect true rhythmic genes.

A concern about the manuscript is that when the authors explain the benchmarking methods, it is not clear whether the methods are benchmarked for their ability to detect rhythmicity or their capacity to predict functionally relevant genes. The methods are built to perform the former and that should be the main criterion to evaluate them. The benchmarking approaches used are correct; however, the concern raised by the authors about identifying non-functionally relevant genes with rhythmic expression patterns is a general concern and not specific for these methods. Thus, this should be made clearer in the text.

In more detail, there are several concepts that need clarification or further discussion:

1) Conservation of rhythmicity and biological relevance.

In several parts of the text, especially in the abstract, the authors seem to use conserved rhythmicity in orthologs as biological/functional relevance and unconserved rhythmicity as biological irrelevance.

For instance the authors write in the abstract: ‘In this study, we show that the nycthemeral rhythmicity at the gene expression level is biologically functional and that this functionality is more conserved between orthologous genes than between random genes’, ‘Rhythmic genes defined with a standard p-value threshold of 0.01 for instance could include genes whose rhythmicity is biologically irrelevant’.

The authors show indeed that rhythmicity is enriched among orthologs, which could indicate that this expression pattern is conserved for functional reasons. This has also been shown in other contexts (i.e. posttranslational modifications – functional sites) and can in principle be used as a way to prioritize these rhythmic genes. However, unconserved rhythmic expression patterns can also be functionally/biologically relevant. The authors mention this in the text: ‘It should be noted that this does not imply that we expect all rhythmic behavior to be conserved between orthologs’. However, this is somehow not reflected and clear in other parts of the manuscript. Even though the way the authors use homology to benchmark the methods is appropriate because a higher number of rhythmic orthologs is expected among true rhythmic genes, there is as well an unknown percentage of true positives among the unconserved rhythmic genes. This should be made clearer in the text. Also, it would very beneficial if the authors could provide some examples of conserved rhythmic genes with known relevant function (i.e. overlap with known cycling genes) and/or examples of clear false positives among the unconserved ones.

2) Use of p-values. The authors compare the overlap of rhythmic genes detected by the different methods and observe that “the same genes seem to be called rhythmic by all methods but the threshold of significance appear(s) inconsistent. These results suggest an issue with the significance of p-value thresholds for the methods… Thus the methods agree on a large number of rhythmic genes, but not necessarily on the order of significance 210 among them”. The studied methods follow different approaches to detect rhythmicity, would the authors expect equal p-value ranking by all methods?

Also, it is not clear whether the ‘issue with the significance of p-values thresholds’ is specifically an issue attributed to the studied methods or if this is more a common misuse of p-values (lower p-value, stronger evidence). Clarifying these points would help the reader, constrain the significance of p-values and at the same time benefit the argument of incorporating biological context.

3) Known Cycling Genes. The authors show the distribution of p-values for known cycling genes for each of the methods; however, it might also be useful to see the number of those genes predicted as rhythmic by each method.

4) Access to the code. It would be very beneficial to have access to the code to reproduce the results or replicate them in other datasets or other methods. Could the authors make the code available in a repository (i.e. GitHub)?

Minor changes

Page 3. Line 85. I would suggest changing the term ‘normal’ to refer to healthy/wild type individuals.

Page 4. Line 126. There is no description of the subset of known cycling genes in the Methods section. The authors should consider including a table with these genes in the Additional material.

Figure 1a. The panel aims at explaining the concept of nycthemeral gene expression but even though it is meant to be a simplification, it is very convoluted and may benefit from a simplified flow and a less synthetic explanation in the figure caption.

Rhythmic genes are enriched in highly expressed genes. The number of detected rhythmic genes is enriched among highly expressed genes in all the benchmarked methods. This is not surprising and the authors clearly state this and provide some possible reasons. However, the statement would benefit from a statistical test showing the enrichment, similarly to how the enrichment of orthologs was demonstrated (page 8).

Figure 2a. Transforming the y-axis of the plot into the -log(default p-value) may help visualize better the enrichment of rhythmic genes among the highly expressed genes compared to the lowly expressed ones.

Figure 3, 4. Use of ‘every’ instead of ‘each’ to refer to all the individual time points registered.

Figure 5a. The use of the Venn diagram is redundant with the Upset plot and could be removed. In general in all panels, the colour scheme seems unnecessary and it just makes the diagrams more difficult to read. The authors should consider that the number of overlaps at all levels (methods) might not be as informative as having for instance the number of common predicted genes by all, by half of the methods or by more than two. This would simplify the visualization and convey a similar message. If the message is how similar the results from different methods are, the authors could consider using a Jaccard index heatmap for instance.

Methods. Missing values. There is no description of how missing values were handled (if any).

Reviewer #3: I enjoyed reading the very well-written manuscript by Laloum and Robinson-Rechave. The authors have performed a benchmarking study of existing methods for detecting diurnal rhythms in gene expression. That might sound run-of-the-mill, but the analysis is especially interesting since it highlights p value distributions and evolutionary conservation. In all, a worthwhile study, definitely mature enough to be reported to the community.

I have no major criticisms, only more minor points:

Noting that the number of time points is crucial for detection power is almost trivial. RNA-Seq may be more sensitive, but I cannot imagine anyone expecting that it should outweigh cutting the number of time points by a factor 3 (Zhang et al.). I suggest shortening the report of those results.

The problem with non-uniform null pvalue distributions in original JTK could be corrected (empJTK). The same correction could be made to RAIN to produce "empRAIN", as suggested by Hutchison and Dinner in the cited bioRxiv report, with as they write "little additional effort". If this is true, authors might carry this out and include the results. Note (also the Editor please) that I'm not requiring this extra work. But the authors should in any case emphasize this in the "Limitations and improvement" section in the discussion, to stimulate such an effort.

Something that should be discussed is amplitude. Amplitude is closely related to function: High amplitudes are to most researchers in the field suggestive of (perhaps) more significant function. For most methods, high amplitudes should (for constant noise level) lead to lower pvalues. In fact, since pvalues are themselves random variables with very broad distributions (broader the lower the amplitude, typically), amplitude may be a strong filtering co-variate with higher potential than orthology to isolate "good" rhythmic transcripts. This has of course been noted earlier, see e.g., the "guidelines" JBR paper from 2017 or the original RAIN paper from 2014.

The broad distribution of pvalues and realizing that they are random variables themselves makes the results in Figure 5 quite expected: Venn diagrams by necessity then give a limited overlap for fixed cutoffs. This is something that was thoroughly discussed in the CMLS review by Lück and Westermark in 2016, which could be cited in relation to the Venn diagrams.

Related, regarding the analysis in Figures 6 and 7. It might be interesting to discuss whether rhythmic orthologs tend to have higher amplitudes, leading to a tendency for lower p values, and thus to detection by more methods.

The reporting of the results in Figures 6 and 7 is a bit of a mess to follow. The reader must be helped better to see the essential point. It is no trivial analysis, but the main text might be shortened to focus on the essentials (mainly Figure 7b). Explaining the analysis thoroughly could be done only in the methods section.

Page 3 close to top: To be fair, original JTK could detect an arbitrary waveform with a slight code modification, as stated in the original JTK paper. The limitation is rather that only this one waveform will be detected (a sine curve by default).

On page 7 3rd row from bottom: "efficieny", is "statistical power" meant?

Reviewer #4: There have been several reviews on this subject.

This manuscript adds to the existing literature

It includes more recent algorithms and, admirably, attempts to move beyond synthetic data (with its inherent biases) in the evaluation.

On these fronts the authors should be commended.

I do, however, have several significant concerns (Some more significant than others ;)

Some of the conclusions are overstated

There seem to be a few statistical issues/inconsistencies

One of recommendations seems wholly unsupported (and I would argue wrong)

Still I suspect that authors can address all my concerns

(1) In ALL P value histograms – you show p values less that 0 (the distribution goes to the left of the green 0, about -.2) and greater than 1. (the distribution goes to ~ 1.2) I am guessing/hoping this is a mislabeling? Did you try to do some weird kernel smoothing of these distributions that didn’t reflect their constrained domains?

(2) The significance thresholds used throughout this manuscript are all very low and not reflective of current best practices (see for example the review by Hughes et al the authors site) In the setting of testing ~ 20,000 genes. A p value threshold of 0.01 is likely to be loaded with false positives. The analysis needs to be re-done with a reasonable threshold (q<=.1?) as including multiple test corrections is both important and standard.

(3) The initial analysis showing that the p-value distributions resulting from the application of some of these methods on randomly generated data -is nonuniform seems important. (A) But in the end your argument that this is important is significantly undercut by your eventual use of the p-values as a simple ranking method. If we are just using the numbers for ranking purposes (which on the face of it – I don’t think I would advocate) then the fact that the p values don’t follow a uniform distribution wouldn’t be a deal breaker. You should either explain this – and consider ALL the methods throughout the review , stick to a reasonable p or q value threshold, or provide some reasoning for both excluding the methods based on distributional properties that you ultimately neglect.(B) More theoretical point – the formal argument is that p values should be uniformly distributed under the “null hypothesis” I am not an expert in all these methods. What are thir null-hypthesis with regard to the temporal pattern? No variation? Gaussian white noise? Autoregressive noise? What is the statistical distribution of your random data? Does it match the null hypotheses for these different methods?

(4) You spend a few pages separating data into high/low expression genes – and then mean/variance normalizing gene expression (within a gene) to test if that would influence the identification of cycling genes. But a quick review shows me that several of these methods (e.g JTK, eJTK, GeneCycle) are rank based. (purely use rank of the sample’s expression of Gene X compared to all other samples). By construction these methods should be completely unaffected by your normalization process (which is what you find). Thus these numerical experiments unnecessarily lengthens the manuscript. It seems like you could quickly explain this and then note that the bias toward High expression is not an artifact of (at least those) methods. It either reflects true biology or a lower signal to noise ratio in lowly expressed gene

(5) Figure 4B Analysis of reducing number of data points and comparison of fewer data points over 2 cycles vs more over 1 cycle. You only show the distributions of p values here. This seems insufficient. You could theoretically be identifying completely different genes (with the reduced data sets) as compared to the full data set. Ie (assuming the full data set is a better representation) – the significant genes in the reduced data set could be all false positives. Similarly when you GeneCycle is not influenced by this comparison – you don’t really know that, You have only shown that it picks out the same # of genes – not the same genes.

(6) You repeatedly find that the baboon data set does not seem to have much overlap with the mouse (or other data sets). In reviewing this study I find that is has very low read depth – complicating any quantitative analysis. You might want to raise this possibility? (up to you)

(7) Final Recommendations “Consider biological replicates as new cycles with one replicate” Perhaps I misunderstand your point - I think this is simply wrong. I can’t find that you supported this (one way or another) in your analysis . More to the point there is a big – real world – difference between euthanizing 3 mice every few hours for a day… and doing 1 mouse for three days. In the first case a single/random unknown influence of the shared environment (e.g. there was a loud noise at 12 pm stopping the mice from eating) would influence all three mice euthanized at that time. Concatenating their data as if they happened in 3 consecutive days would make this appear like a rhythmic behavior (a daily peak near noon). Having 3 days of real data would minimize this risk (unless the noise repeats every day at the same time) There are almost always unknown perturbations – this suggestion is a recipe to make them all appear rhythmic.

(8) Throughout the text you refer to conserved cycling as biologically important. (and non-conserved cycling to be unimportant). I understand that you are trying to use evolutionary conservation. But this is a giant oversimplification. The cycling of a gene – unique to a particular animal – might be very important to that animal. Conserved cycling of a transcript – that has a very long half-life protein product – might be completely physiologically unimportant. This language needs to be cleared up throughout. At best could say something “presumed evolutionary importance” but even this is too strong. I leave it to the authors to find some less loaded terminology

But the terminology should be refined

(9) For figures 6 and 7 I am a bit confused. In these comparisons was the analysis in both species done using the same method. (ie e-JTK defines cyciling in the rat liver – and then e-JTK is used to asses cycling in mouse liver). This seems like the more fair test for each method. But in reading the text it seemed like you used GeneCycle to find (for example) the rhythmic genes in the “benchmark” rat set. and then used all the other method in the lung and looked for overlap. If that is what was done --- it should be redone to be fair (but I doubt it will change the results much). If the “fair” test is the one that was already done – the text should be clarified.

(10) The analysis of conservation – while important – is also overstated.

You postulate (A) “Biologically important cycling should be conserved across species” and then jump to (B)“The statistical test that identifies a property most likely to be conserved is the best test. (B) does not follow from (A) For example A test completely unrelated to cycling could pick out a feature that is more likely to be conserved. This does not make it a good cycling test. Indeed this (I think) what you find when you say the “Naïve test” is better. High expression is actually a more phylogencially conserved feature than cycling per se. More to the point this test is assessing a desirable side feature of a cycling evaluation algorithm – but not its main function. As such this is an important – but secondary issue. Again the discussion needs to be reframed and toned down (Limitations better acknowledged)

(11) No mention is made of the precision of the different methods in describing the particulars of a cycling behavior (e.g the amplitude or period or some other feature depending on how the waveform is described. This is often of great import. Particularly as more recent experiments are trying to assess if there is a change in these parameters with a stimulus. These factors should be considered. It looks like some of the earlier reives did this – but as one of the points of this review is to look at newer methods – it should be redone. This might make a particular differenc with regard to questions of using more replicates (or adding time points) in two a smaller number of cycles – or measuring more cycles (eg. 2 animals every hour for 2 day) vs 2 animals every 4 hours for 4 days)

(12) You try to squeeze a lot on the figures. Perhaps I am just to old - but when printed out I find them to be unreadable.

Text and plots need to be bigger. Perhaps plot a bit less on each figure (if thats what you need to do)

**Have all data underlying the figures and results presented in the manuscript been provided?**

Reviewer #1: Yes

Reviewer #2: Yes

Reviewer #3: Yes

Reviewer #4: Yes

PLOS authors have the option to publish the peer review history of their article (what does this mean?). If published, this will include your full peer review and any attached files.

Reviewer #1: Yes: Alan L Hutchison, MD PhD

Reviewer #2: No

Reviewer #3: No

Reviewer #4: No

---

## [Decision Letter · Decision Letter 1]

9 Jan 2020

Dear Dr Robinson-Rechavi,

Thank you very much for submitting your manuscript, 'Methods detecting rhythmic gene expression are biologically relevant only for strong signal', to PLOS Computational Biology. As with all papers submitted to the journal, yours was fully evaluated by the PLOS Computational Biology editorial team, and in this case, by independent peer reviewers. The reviewers appreciated the attention to an important topic but identified some aspects of the manuscript that should be improved.

We would therefore like to ask you to modify the manuscript according to the review recommendations before we can consider your manuscript for acceptance. Your revisions should address the specific points made by each reviewer and we encourage you to respond to particular issues Please note while forming your response, if your article is accepted, you may have the opportunity to make the peer review history publicly available. The record will include editor decision letters (with reviews) and your responses to reviewer comments. If eligible, we will contact you to opt in or out.raised.

- Supporting Information uploaded as separate files, titled 'Dataset', 'Figure', 'Table', 'Text', 'Protocol', 'Audio', or 'Video'.

We hope to receive your revised manuscript within the next 30 days. If you anticipate any delay in its return, we ask that you let us know the expected resubmission date by email at ploscompbiol@plos.org.

Sincerely,

Attila Csikász-Nagy

Associate Editor

PLOS Computational Biology

Bjoern Peters

Benchmarking Editor

PLOS Computational Biology

[LINK]

Reviewer's Responses to Questions

**Comments to the Authors:**

Reviewer #2: The authors have responded all the comments and edited the manuscript with the appropriate changes.

Reviewer #3: I am still endorsing this paper and am generally happy with the revisions. A careful re-reading revealed the following minor points.

Density estimates for p value distributions go beyond the support for the p values, i.e. have mass at < 0 and > 1. Density estimates can easily be restricted to 0 <= p <= 1, which should be done.

Re: Figure S12b. The main text correctly mentions the lack of consistent p value correlation, but the naive reader may be mislead by the astronomically low p values. This may be related to the KS tests e.g. Figure 5d: the KS test is sensitive to the most minute differences in distributions: even very very similar distributions with minuscule differences are often detected by this test. Thus, some kind of deviance measure to assess similarity of the distributions would be beneficial here. Use of Pearson correlation coefficients: Especially for Figure S12 it seems important to use e.g., a permutation test to assess significance since the log p values cannot be assumed to follow a normal distribution. Was this done?

New text "Of note, the comparison of 260 species under different conditions (light-dark versus dark-dark) is a limitation in itself 261 since the overlap of the rhythmic transcriptome between these two conditions has been 262 shown to be low":

Where has it been shown to be low? In fact I even discourage the use of the term "overlap" which is arbitrary and can be misleading; overlap is usually defined by arbitrary cutoffs. It introduces a black/white thinking into a highly greyscale reality. In any case, mouse liver LD/DD were compared in the DODR paper and the correspondence did not seem "low".

The caption of Figure S5 mentions 7 methods, only 5 are shown.

**Have all data underlying the figures and results presented in the manuscript been provided?**

Reviewer #2: Yes

Reviewer #3: None

PLOS authors have the option to publish the peer review history of their article (what does this mean?). If published, this will include your full peer review and any attached files.

Reviewer #2: No

Reviewer #3: No

---

## [Editor Report · Decision Letter 2]

18 Jan 2020

Dear Prof. Robinson-Rechavi,

We are pleased to inform you that your manuscript 'Methods detecting rhythmic gene expression are biologically relevant only for strong signal' has been provisionally accepted for publication in PLOS Computational Biology.

Before your manuscript can be formally accepted you will need to complete some formatting changes, which you will receive in a follow up email. A member of our team will be in touch within two working days with a set of requests.

Best regards,

Attila Csikász-Nagy

Associate Editor

PLOS Computational Biology

Bjoern Peters

Benchmarking Editor

PLOS Computational Biology

---

## [Editor Report · Acceptance letter]

5 Mar 2020

PCOMPBIOL-D-19-01419R2 

Methods detecting rhythmic gene expression are biologically relevant only for strong signal

Dear Dr Robinson-Rechavi,

I am pleased to inform you that your manuscript has been formally accepted for publication in PLOS Computational Biology. Your manuscript is now with our production department and you will be notified of the publication date in due course.

With kind regards,

Sarah Hammond
